# Inhibition of α(1,6)fucosyltransferase: Effects on Cell Proliferation, Migration, and Adhesion in an SW480/SW620 Syngeneic Colorectal Cancer Model

**DOI:** 10.3390/ijms23158463

**Published:** 2022-07-30

**Authors:** Rubén López-Cortés, Laura Muinelo-Romay, Almudena Fernández-Briera, Emilio Gil-Martín

**Affiliations:** 1Doctoral Program in Methods and Applications in Life Sciences, Faculty of Biology, Campus Lagoas-Marcosende, Universidade de Vigo, 36310 Vigo, Spain; rlcortes.eu@gmail.com; 2Liquid Biopsy Analysis Unit, Translational Medical Oncology (Oncomet), Health Research Institute of Santiago de Compostela (IDIS), CIBERONC, Travesía da Choupana, 15706 Santiago de Compostela, Spain; lmuirom@gmail.com; 3Department of Biochemistry, Genetics and Immunology, Faculty of Biology, Campus Lagoas-Marcosende, Universidade de Vigo, 36310 Vigo, Spain; abriera@uvigo.es

**Keywords:** *FUT8*, core fucosylation, *Lens culinaris* agglutinin, colorectal cancer, proliferation, migration, cancer progression, shRNA

## Abstract

The present study explored the impact of inhibiting α(1,6)fucosylation (core fucosylation) on the functional phenotype of a cellular model of colorectal cancer (CRC) malignization formed by the syngeneic SW480 and SW620 CRC lines. Expression of the *FUT8* gene encoding α(1,6)fucosyltransferase was inhibited in tumor line SW480 by a combination of shRNA-based antisense knockdown and *Lens culinaris* agglutinin (LCA) selection. LCA-resistant clones were subsequently assayed in vitro for proliferation, migration, and adhesion. The α(1,6)FT-inhibited SW480 cells showed enhanced proliferation in adherent conditions, unlike their α(1,6)FT-depleted SW620 counterparts, which displayed reduced proliferation. Under non-adherent conditions, α(1,6)FT-inhibited SW480 cells also showed greater growth capacity than their respective non-targeted control (NTC) cells. However, cell migration decreased in SW480 after *FUT8* knockdown, while adhesion to EA.hy926 cells was significantly enhanced. The reported results indicate that the *FUT8* knockdown strategy with subsequent selection for LCA-resistant clones was effective in greatly reducing α(1,6)FT expression in SW480 and SW620 CRC lines. In addition, α(1,6)FT impairment affected the proliferation, migration, and adhesion of α(1,6)FT-deficient clones SW480 and SW620 in a tumor stage-dependent manner, suggesting that core fucosylation has a dynamic role in the evolution of CRC.

## 1. Introduction

Colorectal cancer (CRC) is the third most common cancer in men and women, as well as the third most prevalent cause of death among adult cancer patients in industrialized countries [1]. While modern surgery and improved radiochemotherapeutic regimens have increased survival rates, CRC remains elusive, because detection in the silent early asymptomatic stages is difficult. Indeed, despite the annual growth of colonoscopic examinations being performed, the compliance of targeted cohorts and, thus, the detection rates among people without clinical symptoms of disease, remain very low; therefore, population-based screening programs typically capture only a small percentage of the overall incidence [2]. Consequently, most diagnoses correspond to advanced cases, for which the prognosis worsens, because almost 50% of operated patients eventually develop recurrent disease and/or metastasis leading to death [3].

CRC is a heterogeneous disease whose malignization and dissemination result from a complex sequence of genetic impacts and phenotype changes [4]. Thus, to improve expected CRC survival, a deeper understanding of the molecular machinery that promotes colorectal tumorigenesis and progression is required. In this regard, it is crucial to identify and characterize the protein drivers that determine the malignant evolution of colorectal cells, and this has attracted great attention. However, despite the intense research developed in the last decades, the cellular and biochemical mechanisms leading to CRC infiltration, spreading, and metastasis are not yet fully understood [4,5].

The enzyme α(1,6)FT, coded by the *FUT8* gene (14q23.3) [6], is the only non-redundant glycosyltrasferase, as well as the only enzyme that specifically transfers GDP-L-fucose via α(1,6) linkage to the innermost N-acetylglucosamine of N-linked glycans [7]. In a previous investigation, our group demonstrated altered α(1,6)FT activity and expression in specimens from CRC-resected patients [8]. The α(1,6)FT activity was found to be increased in tumor tissue compared with healthy tissue, and declined as tumor stage and lymph node infiltration progressed. Likewise, α(1,6)FT activity was enhanced in polypoid tumors, while it also decreased as the degree of infiltration progressed. Subsequent research has reported that α(1,6)FT expression is a good predictor of local disease recurrence, and therefore an accurate indicator of poor prognosis in CRC [9]. Based on the fact that altered protein glycosylation profoundly influences cancer phenotype, we next examined α(1,6)fucosylated proteins differentially expressed in CRC as potential candidates for drivers of CRC malignancy [10]. The findings were consistent with previous reports on lung cancer [11], hepatocarcinoma [12,13], pancreatic cancer [14], and gastric cancer [15], in which the expression of α(1,6)FT has been closely related to cancer prognosis. However, the important question of whether α(1,6)FT works in the same way, regardless of the tumor stage or the histological origin, remains open.

To gain insight into the manner by which α(1,6)FT can modulate CRC outcomes, we analyzed the impact of inhibiting the enzyme’s expression in the isogenic SW480 and SW620 CRC lines through knockdown of their *FUT8* gene. This cellular tandem came from two consecutive pathogenic steps involving a colorectal tumor removed from a 50-year-old Caucasian male [16]. The epithelial-like non-metastatic SW480 line was generated from grade 3–4 primary colon adenocarcinoma, while the metastatic fibroblast-like SW620 line was established several months later from a lymph node of the same tumor after it metastasized. These two lineages provided a cellular model of CRC malignization and progression [17,18,19] that could be interrogated in vitro to screen the impact of α(1,6)FT downregulation on the CRC phenotype. Indeed, this approach allowed us to reveal the impact of *FUT8* knockdown on epithelial–mesenchymal transition (EMT) markers, as well as on the proliferation, migration, and adhesion of SW480 and SW620 cells, suggesting that core fucosylation may play a dynamic role in the malignant evolution of CRC.

## 2. Results

### 2.1. FUT8 Knockdown Altered Epithelial–Mesenchymal Transition (EMT) Markers in SW480 and SW620 Cell Lines

α(1,6)FT expression was first downregulated using an shRNA interference-based approach. To this end, the SW480 and SW620 CRC lines were transfected with lentiviral particles carrying five different anti-human *FUT8*-gene shRNAs (Appendix A). Of the five transfected clones obtained from each tumor line, we selected those with the highest knockdown at the mRNA and protein levels (the SW480/SW620 F52 and SW480/SW620 F59 clones; Appendix A). Subsequently, we analyzed the expression of EMT markers in the SW480 and SW620 cells with the *FUT8* gene silenced (Figure 1 and Figure 2, respectively). The SW480 knockdown cells (the SW480 F52 and F59 clones) showed reduced E-cadherin expression in comparison with the non-targeted transfected control (SW480 NTC; Figure 1A,C). On the contrary, in the SW620 cells, the F59 knockdown clone showed increased E-cadherin (Figure 2A,C) and N-cadherin (Figure 2A,B) levels. Vimentin was increased in the SW480 F52 clone (Figure 1A,D), while it was not expressed in the SW620 knockdown clones (Figure 2A). Regarding β-catenin (Figure 1A,E and Figure 2A,E) and c-Kit protein (Figure 1A,F and Figure 2A,D), the silenced F52 and F59 clones of both CRC lines showed higher levels than the non-targeted controls.

### 2.2. LCA Treatment of shRNA-Transfected SW480 and SW620 Clones Reinforced Depletion of Core Fucosylation

In order to further decrease the α(1,6)fucosylation, the *FUT8* knockdown clones were treated with LCA for 1 week. LCA is a plant lectin with high specificity for binding to mono- and bi-antennary *N-linked,* core-fucosylated *oligosaccharides* [20], and it has antiproliferative capacity in certain tumor cells, in which it triggers cell death response [21]. Therefore, LCA exposure can screen cell populations with exhausted core fucolsylation levels. In this regard, the presence of LCA in the culture medium yielded significant proliferative differences between the *FUT8* knockdown clones (SW480 F52L, SW480 F59L, SW620 F52L, and SW620 F59L) and their transduction controls (SW480 NTC and SW620 NTC). Specifically, after 24 h of lectin treatment, the α(1,6)FT-attenuated clones survived and propagated in the LCA-containing medium, while their NTC counterparts hardly proliferated (Figure 3A,B). Even at the lowest LCA concentrations, the NTC cell populations were reduced below 10% of the initial seeding number (data not shown). By contrast, SW480 and SW620 knockdown clones developed lectin resistance at the end of treatment, and reached confluency long after 120 h of LCA exposure. However, a difference was observed in the viability of the LCA-resistant SW480 cells. The SW480 F59L clone was significantly more sensitive than the SW480 F52L clone to the proapoptotic effect of the LCA (*p* < 0.05; Figure 3A), even though both showed similar inhibition of *FUT8* mRNA and α(1,6)FT expression (Figure 3C,D and Figure 3E,F, respectively).

It should be noted that the LCA selection greatly increased the downregulation of core fucosylation in the knockdown SW480 and SW620 cells (Appendix A). This observation confirmed that the combination of *FUT8* knockdown and LCA selection downregulates α(1,6)FT expression; thus, we investigated the functional impact of inhibiting core fucosylation in CRC.

### 2.3. α(1,6)FT Downregulation Showed Opposite Effect on Proliferation of SW480 and SW620 Cells

Proliferation is a key process in the growth and colonization of CRC. Correspondingly, the effect of *FUT8* knockdown on the proliferation capacity of SW480 and SW620 LCA-resistant clones was investigated. Under adherent conditions, SW480 knockdown clones displayed statistically significantly increased proliferation rates (*p* < 0.05 according to unpaired *t*-test) compared to the NTC clone (Figure 4A), whereas in SW620 the effect was the reverse (*p* < 0.05 according to unpaired *t*-test; Figure 4B). Similarly, in an anchorage-dependent context (soft agar), the colony-forming capacity of the SW480 F52L clone was significantly enhanced (*p* < 0.05 according to unpaired *t*-test; Figure 4C), while the F52L/F59L SW620 clones showed proliferation rates equivalent to their non-targeted controls.

### 2.4. α(1,6)FT Downregulation Altered Cell Migration and Adhesion of SW480 Cell Line

Further important tumor cell spreading abilities are: (i) migration, which is essential for detaching from the primary tumor; (ii) cell–cell/cell–matrix adhesive interactions, which are pivotal for binding to endothelial cells and achieving intravasation/extravasation for metastatic dissemination. In this regard, we analyzed the impact of *FUT8* knockdown on the adhesion and migration potential of α(1,6)FT-deficient SW480 and SW620 cells. In Boyden chamber assays, the SW480 knockdown clones showed significantly lower migration rates than their NTC counterparts (*p* < 0.05 according to unpaired t-test for the SW480 F52L clone; Figure 5A), while the SW620 knockdown clones showed no significant changes relative to their non-targeted controls (Figure 5B). Moreover, the SW620 cells defective in α(1,6)FT showed lower migration capacity than their SW480 counterparts (Figure 5B vs. Figure 5A). Similar migration results were obtained by wound-healing assays (Appendix A).

The substrate adherence to different matrices of *FUT8*-silenced clones was assessed using non-treated, collagen-coated, and endothelium-covered culture plates (Figure 5C,D). Cell adhesion to non-coated control plates was similar in all the analyzed SW480 and SW620. Adherence to collagen-coated plates was examined, to investigate the capacity for extracellular interaction and integrin-mediated cell attachment. In this regard, a trend of reduced adhesion on the collagen type I-coated plates of the SW480 and SW620 *FUT8* knockdown clones was determined, although the slight differences observed were not significant (Figure 5C,D). By contrast, adhesion to epithelial Ea.hy926 cells showed that the SW480 knockdown clones had significantly higher attachment capacity than the respective non-targeted controls (*p* < 0.05 according to unpaired *t*-test; Figure 5C). In the case of the SW620 knockdown clones, no differences were reported (Figure 5D).

## 3. Discussion

Core-α(1,6)fucose plays a relevant role in cell homeostasis, as revealed by its neonatal lethality in *FUT8*-null mice [22,23], and the phenotypic complications of individuals harboring pathogenic variants of *FUT8* [24,25]. Cancer is not an exception and, as a result, core fucosylation disorders have proven to be relevant in the pathogenesis and clinical outcome of cancer patients [26,27]. In this regard, our group reported increased α(1,6)FT activity and expression in the tumor tissue of CRC specimens [8], as well as their potential as a prognostic factor [9]. Specifically, we concluded that core fucosylation describes an oscillating evolution, as the malignant potential of the tumor enhances; tumoral α(1,6)FT activity is initially upregulated in comparison with healthy tissue and preneoplastic colorectal lesions, while it progressively decays as the cancer becomes more aggressive [8]. Regarding this, we established that overexpression of guanosine 5′-diphosphate (GDP)-L-fucose transporter (GDP-Fuc-Tr) in the tumor tissue may be a contributing factor to the upregulation of α(1,6)fucosylation in CRC patients [28]. Furthermore, we proved that α(1,6)FT expression can be a plausible predictor of poor prognosis in this neoplasia [9], and identified differentially expressed α(1,6)fucosylated proteins in tumor tissue, that could potentially be candidate biomarkers [10].

In order to determine the still-not-understood role of α(1,6)FT during carcinogenesis and dissemination of CRC, the expression of the enzyme was reduced, in the present study, by knockdown of the *FUT8* gene in a model of malignancy formed by the syngeneic SW480 (premetastatic) and SW620 (metastatic) tumor lines [17]. This approach has allowed us to report that the reduction of α(1,6)FT had a greater impact on colorectal cell functionality in the premetastatic stage than in later phases of the tumor cycle, suggesting the potential of the enzyme to modulate tumor progression.

This background indicated the possibility of obtaining fully core-fucosylated glycoproteins, despite removing α(1,6)FT by different interference *FUT8-*downregulation approaches, including single siRNA [29] and double siRNA [30], as well as more recent editing alternatives through zinc finger nucleases [31] or CRISPR/Cas9 [32]. It is also known that exposure to LCA is crucial in the production of therapeutic glycoengineered fucose-free antibodies [29], and that mutants expressing different glycosylation patterns display varying degrees of resistance to LCA [33]. Based on this evidence, we proceeded to an LCA-mediated selection of *FUT8* knockdown cells, thus achieving the maximum reduction of core fucosylation in lectin-resistant clones (Figure 3).

After knockdown of the *FUT8* gene, SW480 and SW620 clones deficient in core fucosylation were analyzed for the expression of some important proteins related to EMT, as α(1,6)fucosylation impairment has been implicated in kidney, lung, and breast cancer [34,35,36]. The panel of EMT mediators included the epithelial marker E-cadherin, the mesenchymal-associated proteins vimentin and N-cadherin, the receptor tyrosine kinase c-Kit, and a core component of the Wnt signaling cascade, β-catenin. The results from the assessment of these EMT mediators showed a different impact of *FUT8* knockdown on SW480 and SW620 lines. In SW620 cells deficient in α(1,6)FT expression, the main finding was complete downregulation of type III intermediate filament vimentin (Figure 2A), which is frequently overexpressed in CRC and associated with invasiveness and poor prognosis [37]. On the other hand, α(1,6)FT-deficient SW480 cells (F52 clone) showed reduced E-cadherin and enhanced vimentin levels (Figure 1C,D). In this regard, the increased vimentin/E-cadherin ratio has been proposed as a hallmark of EMT changes in CRC cells [38], which may help in monitoring malignancy [39] and patient prognosis [40]. Moreover, we reported that *FUT8*-silencing increased the stem cell factor (SCF)-activated tyrosine kinase receptor c-Kit in SW480 cells (Figure 1F). The elevated c-kit signaling, which promotes proliferation and invasiveness potential in CRC [41,42], may be evidence of the feed-forward that EMT might undergo in SW480 clones after the shutdown of core fucosylation. The knockdown of *FUT8* also increased the intracellular Wnt pathway transducer β-catenin in SW480 (Figure 1E) and SW620 (Figure 2E) cells. In this regard, the absence of core fucose in embryonic fibroblasts due to *FUT8* downregulation was previously correlated with the upregulation of Wnt/β-catenin signaling [43], reinforcing our current data. Overall, the profiling of EMT markers suggests that α(1,6)FT suppression favors the acquisition of a more mesenchymal phenotype in the premetastatic SW480 cell line, specifically demonstrated by the increased vimentin/E-cadherin ratio. However, EMT is a non-uniform, multi-step process that proceeds through intermediate epithelial–mesenchymal states displaying dynamic changes in their downstream regulators [44,45]; hence, transdifferentiation of epithelial cells should be evaluated by taking into account the evolution of phenotype features, instead of relying exclusively on biochemical changes [46]. Henceforth, a more in-depth characterization of α(1,6)FT in the different stages of CRC is required, to understand how core fucosylation influences EMT drivers and may contribute to the carcinogenic process.

After LCA selection had reinforced the downregulation of α(1,6)FT at mRNA and protein levels, we addressed the functional phenotype of core fucosylation-deficient cells, through in vitro assays that mimicked important characteristics of tumor growth and dissemination. We first studied cell proliferation in adherent and non-adherent growth conditions. In an adherent medium, *FUT8*-silenced SW480 clones showed significantly increased proliferation, while in their SW620 counterparts it was significantly reduced (Figure 4A,B). Such divergent responses may have come from the different regulation of proliferative factors in both lines. In this regard, the proliferation-promoting c-kit factor, normally overexpressed in colorectal tumors with greater proliferative and invasive potential [47], was found to be upregulated in SW480 *FUT8*-deficient clones, and minimally altered in SW620 clones (Figure 1F and Figure 2D). The overproliferation observed in α(1,6)FT-deficient SW480 clones was consistent with the mesenchymal profile, indicated by their increased vimentin/E-cadherin ratio (Figure 1C,D). On the other hand, *FUT8*-silenced SW480 cells grew significantly better on agarose than they did under adherent conditions. Anchorage-independent growth is a hallmark of anoikis resistance and metastasis [48]. In this sense, only SW480 cells have been described as sensible to anoikis, unlike their metastatic SW620 counterparts, which become anoikis-resistant during tumor progression [49]. Consequently, the suggested increase in the malignant potential of the SW480 line caused by suppression of core fucosylation could also include the loss of sensitivity to cell death signals.

Results from transwell Boyden chamber assays showed that α(1,6)FT inhibition impaired the migration capacity of the SW480 line (Figure 5A). The reduced invasiveness caused by *FUT8* knockdown in SW480 cells has been described previously in melanoma, pancreatic, lung, and breast cancer, among others [35,36,50,51]. Furthermore, in lung and breast cancer, the role of α(1,6)FT in cell migration has mainly been associated with the inhibition of EMT induced by TGF-β [35,36]. However, based on the scope of our results, we cannot specifically associate the reduced migration capacity observed in *FUT8* knockdown SW480 cells with EMT impairment, as previously discussed. Instead, other indirect mechanisms are possible. In melanoma cells, for example, α(1,6)FT inhibition is associated with neural cell adhesion molecule L1 (L1CAM) cleavage, and the ability of this molecule to promote melanoma invasion [51,52]. Alternatively, in *FUT8*^–/–^ embryonic murine fibroblasts, the almost complete lack of integrin-α3/β1 core fucosylation led to highly reduced migration [53]. In addition, *FUT8* interference reduced the expression of metalloproteinase 2, and especially metalloproteinase 9, in breast cancer cells [54], impairing the extracellular matrix reorganization required for tumor spread.

Finally, α(1,6)FT-deficient SW480/SW620 clones were tested for adhesion (Figure 5C,D), to address the relevance of suppressing core fucosylation in CRC metastasis. At this point, we observed a significant increase in the adhesion of α(1,6)FT-deficient SW480 clones to Ea.hy926 epithelial cells, suggesting that improved endothelial–epithelial interactions might enhance carcinogenicity in cells downregulated for core fucosylation. It is important to note that endothelial–epithelial interactions are associated with membrane adhesion proteins, including α(1,6)fucosylated E-cadherin [55], whose expression and activity have been described as being modulated by core fucosylation in CRC [41], lung [56], and breast [54] cells. Furthermore, the interaction of *α*v*β*3 integrin with L1-CAM and PECAM-1 is relevant for trans-endothelial dissemination of CRC cells through adhesion to endothelial cells [57]. These membrane proteins are targeted by α(1,6)FT, and are thus hypothesized to be mediators of the effects observed in SW480 cells after *FUT8* knockdown.

Taken together, the results of α(1,6)FT downregulation in the tandem of syngeneic SW480 and SW620 CRC lines indicate that core fucosylation modulates EMT and some phenotype features involved in carcinogenesis and metastasis of colorectal cells.

## 4. Materials and Methods

### 4.1. Cell Culture

The syngeneic SW480 and SW620 CRC lines obtained from the American Type Culture Collection (ATCC; Manassas, VA, USA), were kindly donated by the Health Research Institute of Santiago de Compostela (IDIS, Spain). Wild-type cells were maintained in high-glucose Dulbecco’s modified eagle medium (DMEM; Sigma–Aldrich, St. Louis, MO, USA), supplemented with 10% fetal bovine serum (FBS; Life Technologies, Grand Island, NY, USA) and 10,000 U/mL penicillin–streptomycin (Life Technologies, Grand Island, NY, USA) at 37 °C in a humidified incubator supplied with 5% CO_2_. Knockdown and non-targeted control (NTC) cells were also supplemented with 5 μg/mL puromycin (Sigma–Aldrich, St. Louis, MO, USA).

### 4.2. Protein Extraction from Cell Culture

Cell lysates were obtained by direct addition of radioimmunoprecipitation assay (RIPA) buffer (0.1% SDS, 150 mM NaCl, 50 mM Tris-HCl, pH 8.5, 0.5% sodium deoxycholate, 1% Nonidet P-40, 2 mM Na_3_VO_4_, 4 mM NaF) supplemented immediately before use with complete protease inhibitor cocktail tablets (Roche Life Sciences, Penzberg, Bayern, Germany). Cell lysis was performed by hand, using a scraper. Thereafter, the suspension was kept on ice for 30 min, with soft vortexing at intervals. Cell debris and non-solubilized material were removed by centrifugation at 10,000× *g* (10 min, 4 °C), and stored at −20 °C. Protein content was analyzed by the bicinchoninic acid (BCA) method (Sigma–Aldrich, St. Louis, MO, USA), using bovine serum albumin (BSA; Sigma–Aldrich, St. Louis, MO, USA) as reference to define the calibration curve.

### 4.3. SDS-PAGE

Next, 10 µg of protein per sample was heated in 4× Laemmli loading buffer for 5 min at 100 °C, and separated by SDS-PAGE in 10% polyacrylamide gels. The voltage was set at 200 V for 50 min, or until the blue front eluted. Protein bands were completely resolved in the 260–20 kDa range with adequate standard molecular weight markers (BlueStar PLUS Prestained Protein Marker, Nippon Genetics Europe GmbH, Düren, Nordrhein-Westfalen, Germany). Heavier proteins required the use of 12% polyacrylamide gels, instead of 10%. Coomassie brilliant blue (0.1% Coomassie blue R250, 5% glacial acetic acid, 30% methanol) was used for on-gel protein staining.

### 4.4. Immunoblot and Lectin Blot

Gels were blotted onto polyvinylidenedifluoride (PVDF) membranes. After blotting, the membranes were blocked for 1 h with 5% skimmed milk in Tris-buffered saline (TBS; 20 mM Tris, 150 mM NaCl, pH 7.4). Next, the membranes were incubated for 1 h at room temperature with a primary antibody, 1/500 of mouse monoclonal anti-*FUT8* IgG (Proteintech, Chicago, IL, USA) in TBST (0.05% Tween 20 in TBS). Developing was performed by incubating with Clarity Western ECL detection reagent (Bio-Rad, Hercules, CA, USA), in accordance with the manufacturer’s protocol.

For lectin blot, the membranes were blocked for 1 h with 3% BSA in TBS. The specific fucose biotinylated *Pholiota squarrosa* lectin (PhoSl; chemically synthesized by Peptide 2.0 Inc., Chantilly, VA, USA) and *Lens culinaris* agglutinin (LCA; Vector Laboratories, Peterborough, UK) were sequentially incubated in TBST at 1/350 and 1/2000, respectively. An intermediate mild stripping step (15 g/L glycine, 1 g/L SDS, 0.01% Tween 20, pH 2.2) was inserted between the lectin blots. Subsequently, the membranes were incubated for 1 h with horseradish peroxidase-conjugated avidin (VECTASTAIN Elite ABC Kit; Vector Laboratories, Peterborough, UK). Chemiluminescence was developed by incubating with Clarity Western ECL detection reagent, in accordance with the manufacturer’s instructions. The intensity of staining was quantified by using Fiji software [58], and plotted with Microsoft Excel 2013. The statistical analysis was carried out with IBM SPSS Statistics v22.

### 4.5. shRNA Lentiviral Transfection

Next, 96-well plates containing 1.6 × 10^4^ cells were freshly seeded and incubated overnight in a humidified incubator at 37 °C and 5% CO_2_ atmosphere. Lentiviral particles containing pLKO.1 plasmids targeting human *FUT8* (MISSION Lentiviral Transduction Particles pLKO.1-puro-CMV-TurboGFP, TRCN0000035952, TRCN0000035953, TRCN00000229959, TRCN00000229960, and TRCN00000229961) and a non-targeting control (MISSION^®^ pLKO.1-puro-CMV-TurboGFP, SHC003) were purchased from Sigma–Aldrich (St. Louis, MO, USA) (Appendix A). Briefly, 110 μL/well of hexadimethrine bromide (Sigma–Aldrich, St. Louis, MO, USA), at a final concentration of 8 μg/mL, and 15 μL/well of lentiviral particles were added ab initio. The cells were then incubated overnight, to allow transfection. Afterwards, DMEM was removed and replaced with fresh DMEM containing 5 μg/mL of puromycin. This medium was replaced every 72 h until resistant clones grew.

### 4.6. Selection and Phenotypic Analysis of FUT8 Knockdown Cells

Phenotypic selection of *FUT8* knockdown clones was achieved by long exposure to LCA, which binds the α(1,6)fucosylated structures of N-linked oligosaccharides [59], and forces the cells expressing this structural motif to the apoptotic pathway [60,61]. LCA selection was initiated by supplementing the complete medium with 500 μg/mL of LCA from a 5 mg/mL LCA stock solution. Clones were grown in an LCA-containing medium for 7 days. Afterwards, the cells were seeded in a complete medium without LCA.

### 4.7. RNA Extraction and Quantitative Real-Time PCR (RT-qPCR)

Total mRNA from the cell cultures was extracted using the High Pure RNA Isolation Kit (Roche, Applied Science, Indianapolis, IN, USA), and cDNA synthesis was performed with a MuLV Reverse Transcriptase Kit (Applied Biosystems, Foster City, CA, USA), both in accordance with the manufacturer’s indications. RT-q-PCR was carried out by means of TaqMan assays in an ABI 7500 Real-Time PCR System. Glyceraldehyde-3-phosphate dehydrogenase (*GAPDH*) was used as an internal normalization control. Thus, the results were represented as the fold change in gene expression relative to *GAPDH* gene expression (2^−ΔΔCt^).

### 4.8. Cell Proliferation Assay

Experiments on cell proliferation (n = 3) were carried out using alamarBlue (AB; ThermoFisher Scientific, Waltham, MA, USA) as a developing reagent. The cells were trypsinized from sub-confluent cultures, and suspended in a culture medium containing 10% FBS. An initial amount of 5 × 10^3^ cells was seeded into triplicate wells on a 96-well plate (final volume 100 µL/well) under standard culture conditions (37 °C, 5% CO_2_). After 48 h of proliferation, 10 µL of AB was directly added to the culture medium at a final concentration of 10%, and the plate was returned to the incubator for 3 h. The emission of fluorescence was then recorded on a FLUOstar OPTIMA fluorometer plate reader (BMG Labtech, Ortenberg, Germany), with the wavelength filters set at 544 nm for excitation and 590 nm for emission. Medium with AB and without cells was used as blank control.

### 4.9. Colony Formation Assay

Colony formation assays were performed in 96-well plates by seeding 4 × 10^3^ cells resuspended in 100 μL of agar medium solution (0.3%) on top of a more concentrated (0.6%) agar medium layer. Thereafter, cells were incubated for 48 h at 37 °C and 5% CO_2_. After 24 h of proliferation, 10 µL of AB was directly added to the culture medium at a final concentration of 10%, and the plate was returned to the incubator for 1 h. The emission of fluorescence was analyzed every 24 h in the FLUOstar OPTIMA fluorometer, with the wavelength filters set at 544 nm for excitation and 590 nm for emission. For the proliferation assay, AB added to the culture medium without the presence of cells was used as blank control (n = 3).

### 4.10. Transwell Migration Assay

Migration assays were performed with 24-well transwell plates (8.0 µm pore size polycarbonate membrane; Corning, NY, USA). Briefly, 1 × 10^6^ cells were suspended in 200 µL of serum-free medium, and seeded on the upper side of the Boyden chamber. The lower side of the chamber was filled with supplemented medium containing 10% FBS. After incubation at 37 °C for 48 h, non-migrating cells were removed with a cotton swab, and those that migrated through the chamber were trypsinized, collected, stained with 4 µM calcein acetoxymethyl ester (Invitrogen, Paisley, UK) in accordance with the manufacturer’s protocol, and visualized at 485 nm in the FLUOstar OPTIMA fluorometer. Three independent experiments were performed in triplicate. The results are presented as relative percentage of migration.

### 4.11. Adhesion Assay

For adhesion experiments, 3 types of 96-well plates were employed: non-coated; previously coated overnight at 4 °C with Collagen I from rat tail (ThermoFisher Scientific, Waltham, MA, USA); or a monolayer of Ea.hy 926 lung cancer cells. The CRC cells were pre-stained with 4 µM calcein acetoxymethyl ester (Invitrogen, Paisley, UK) in accordance with the manufacturer’s protocol. Afterwards, 5 × 10^5^ cells/well were seeded and incubated at 37 °C for 60 min. Non-adherent cells were removed by gentle washing with PBS, and adherent cells were counted by fluorescence at 485 nm (normalized to the basal point fluorescence intensity) using the FLUOstar OPTIMA fluorometer. The average number of cells (mean ± SD) was calculated based on triplicate experiments.

### 4.12. Wound Healing

Cells were plated in 24-well plates (at a density of 125,000 cells/well), and incubated in a complete medium for 24 h. A pipette tip was used to scratch wounds in the confluent monolayer cells. The cells were then incubated in serum-free medium, and photographed at 0, 24, and 48 h using an inverted microscope. The images were analyzed by Fiji image processing software [58]. The migration ability was expressed as migration percentage (%) = [(cell-free area at 0 h (CFA0)–cell-free area at 24 or 48 h (CFA24 or CFA48))/cell-free area at 0 h (CFA0)] × 100.

### 4.13. Statistical Analysis

GraphPad Prism v9 and IBM SPSS Statistics v22 were used for statistical analysis. An unpaired *t*-test was used to determine the functional differences between *FUT8* knockdown cells and their non-targeted controls. One-way ANOVA and Fisher’s test were used to analyze the effect of LCA treatment on the SW480 and SW620 *FUT8* knockdown clones. In all comparisons, statistical significance was set at *p* ≤ 0.05.

## 5. Conclusions

Downregulation of the core fucosylation enzyme α(1,6)FT enhanced proliferation and cell–cell adhesion in *FUT8* knockdown SW480 clones, while reducing their migration. In contrast, SW620 counterparts deficient in α(1,6)FT showed faint changes in their functional phenotype. High-throughput DNA screening and imaging technologies have revealed different genetic programs and phenotypic traits between primary SW480 cells and isogenic metastatic SW620 cells [62,63]. However, the outlook is far from clear, probably due to the intrinsic heterogeneity of these tumor cells, as two specialized SW480 cell subpopulations have been characterized based on their biophysical properties and involvement in the growth and dissemination of colorectal tumors [62,64]. Nevertheless, our results indicate the importance of core fucosylation in the development and evolution of CRC, and highlight the ongoing interest in elucidating α(1,6)FT and N-glycan dysregulation throughout the tumor cycle [65,66]. In this regard, research on CRC has revealed specific N-glycan signatures in all tumor stages [67], as well as core-fucosylated motifs [68] which, as in other neoplasias, could discriminate primary tumors from metastases based on N-glycomic differences [54,69]. Hence, research to elucidate the role that this enzyme plays in the pathogenesis of CRC is of great translational importance, since strategic advances in new therapies could be based on core-fucosylated targets.

## Figures and Tables

**Figure 1 ijms-23-08463-f001:**
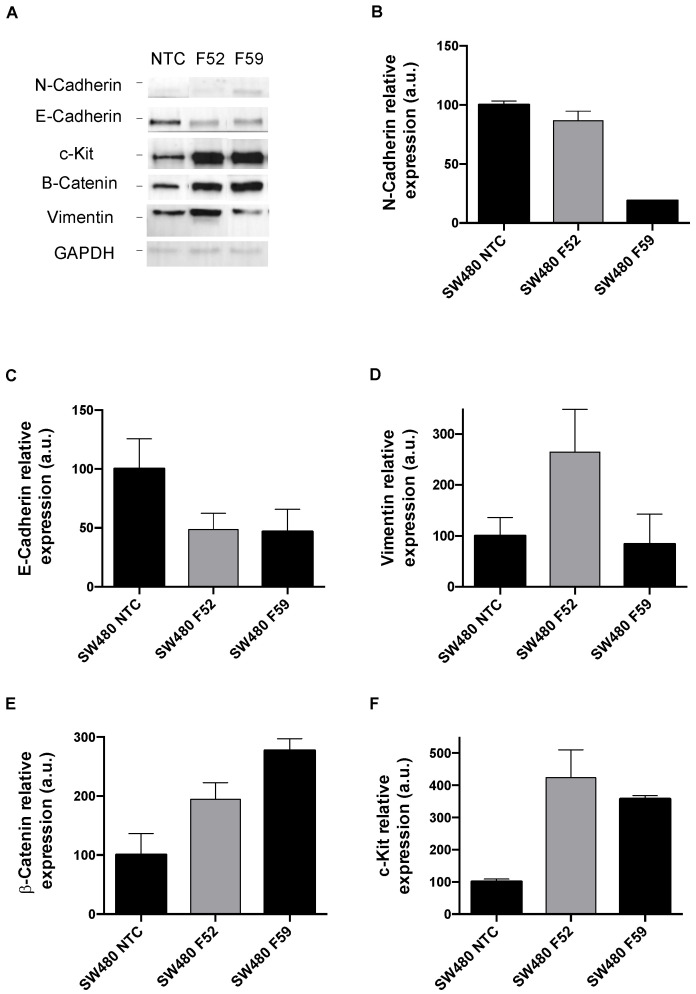
Impact of *FUT8* downregulation on EMT markers in the SW480 cell line. (**A**): Representative image of protein levels of EMT markers analyzed by Western blot in SW480 NTC and *FUT8*-silenced clones F52 and 59. (**B**–**F**): Relative quantification of N-cadherin, E-cadherin, c-Kit, β-catenin, and vimentin using GAPDH protein expression as loading control. Measurements were plotted as mean ± SEM of two replicates. NTC clones were used as reference for statistical analysis.

**Figure 2 ijms-23-08463-f002:**
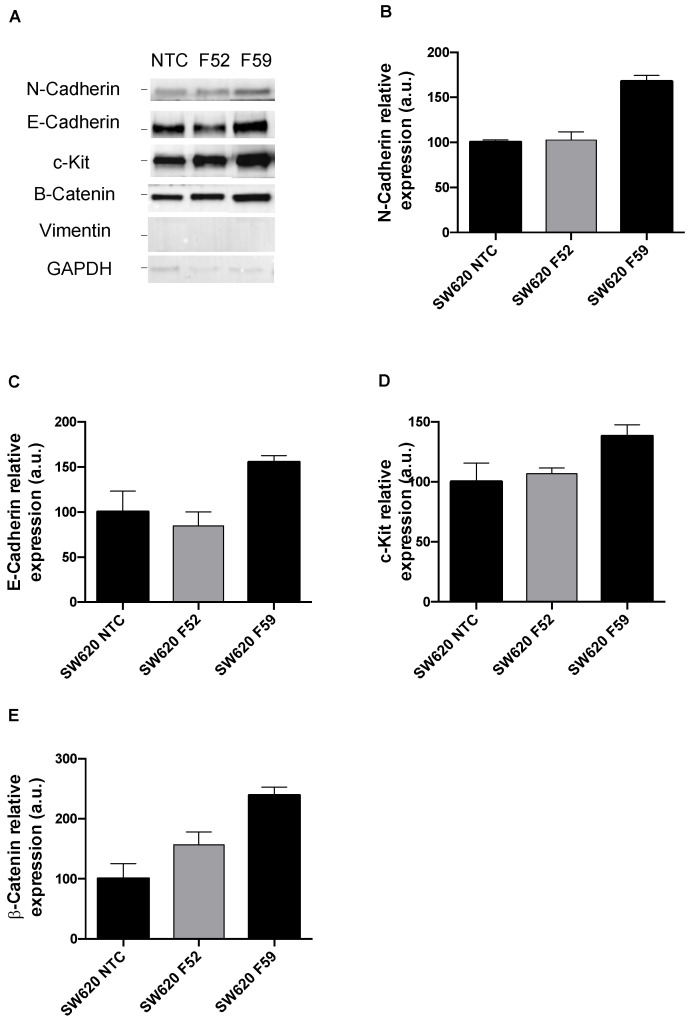
Impact of *FUT8* downregulation on EMT markers in SW620 cell line. (**A**): Representative image of protein levels of EMT markers analyzed by Western blot in SW620 NTC and *FUT8*-silenced clones F52 and 59. (**B**–**E**): Relative quantification of N-cadherin, E-cadherin, c-Kit, β-catenin, and vimentin using GAPDH protein expression as loading control. Measurements were plotted as mean ± SEM of two replicates. NTC clones were used as reference for statistical analysis.

**Figure 3 ijms-23-08463-f003:**
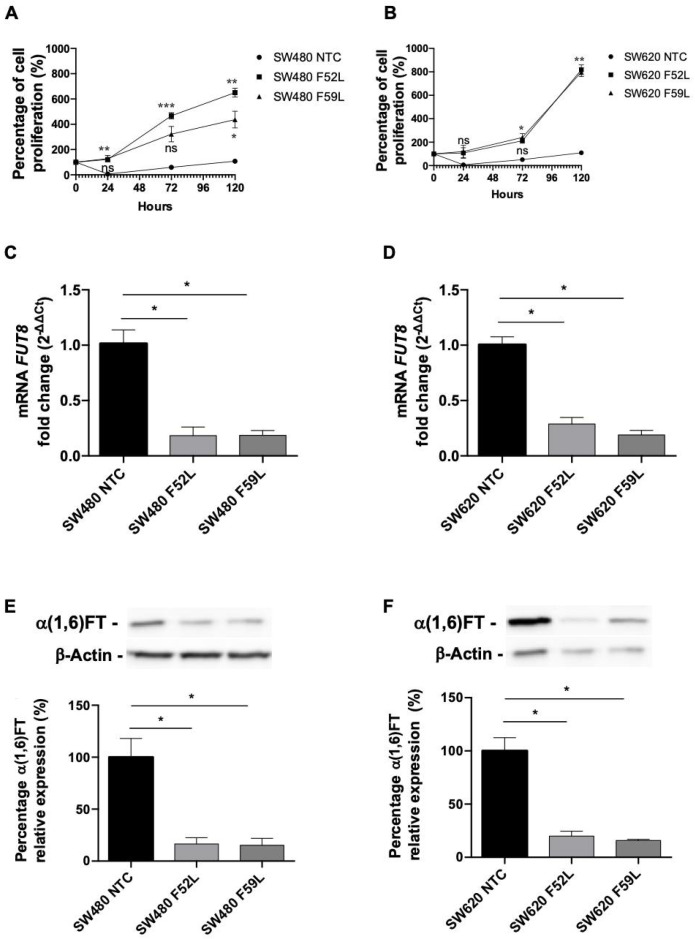
Effect of LCA treatment on SW480 and SW620 *FUT8* knockdown clones. (**A**,**B**): Cell growth dynamics of (**A**) shRNA *FUT8* clones of SW480 and (**B**) SW620 treated with 500 µg/mL of LCA in culture medium. Black circles: non-targeted control (NTC) cells; black squares: shRNA *FUT8* F52L cells; black triangles: shRNA *FUT8* F59L cells. Cell proliferation was measured in triplicate using alamarBlue assay. Results are given as percentage of proliferation, considering basal point for each cell line as 100%. Plots represent mean ± SEM. NTC clones were used as reference for statistical calculations. One-way ANOVA was significant, and multiple comparisons among NTC, F52, and F59 groups were done using Fisher’s multiple comparisons test; * *p* < 0.05; ** *p* < 0.01; *** *p* < 0.001. (**C**,**D**): mRNA *FUT8* levels analyzed by (**C**) RT-qPCR in SW480 and (**D**) SW620 cell models after LCA phenotypic subclone selection. mRNA *FUT8* levels were quantified using mRNA *GAPDH* as housekeeping, and calculated as ΔCt. Relative expression between NTC and silenced clones was determined using 2^−ΔΔCt^ fold change method. Measurements were obtained from three experiments, and results were plotted as mean ± SEM. (**E**,**F**): After lectin selection, Western blot of α(1,6)FT enzyme was performed in (**E**) SW480 and (**F**) SW620 LCA-resistant clones. Measurements were plotted as mean ± SEM of three assays. NTC clones were used for statistical calculations. An unpaired t-test was applied for (**C**,**D**) mRNA and (**E**,**F**) α(1,6)FT expression analysis; * *p* < 0.05.

**Figure 4 ijms-23-08463-f004:**
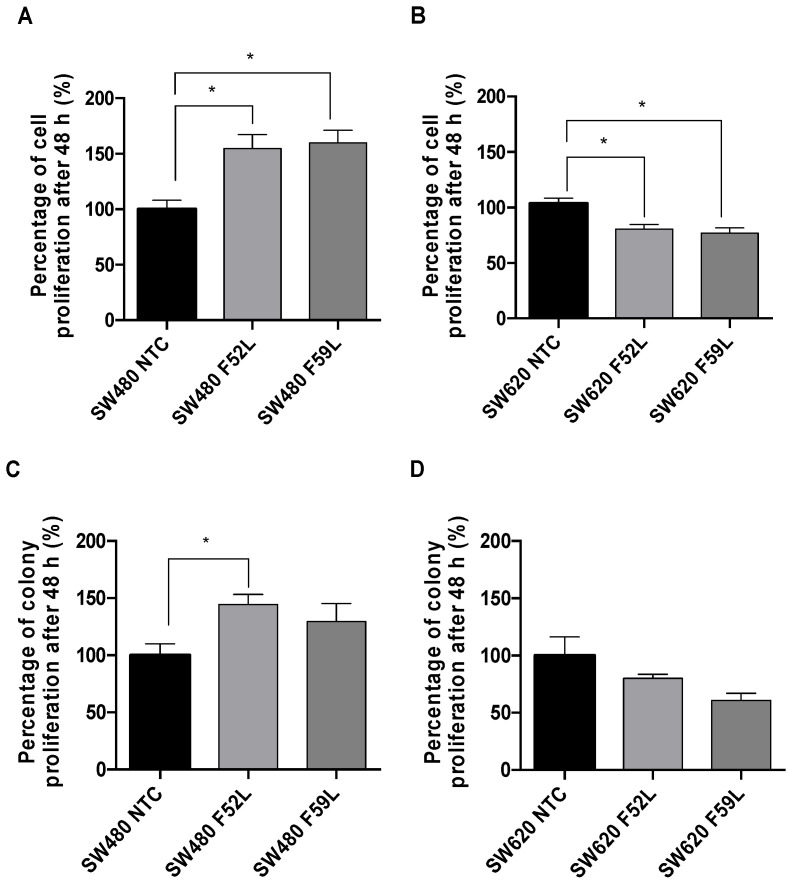
Impact of *FUT8* knockdown on the proliferative activity of SW480 and SW620 cell lines. Cell proliferation in (**A**,**B**) adherent and (**C**,**D**) non-adherent conditions was measured in SW480 and SW620 NTC clones and their corresponding *FUT8-*silenced F52L and F59L clones, using alamar Blue assay. To generate non-adherent conditions, the cells were cultured in soft agar. The results are given as a percentage of proliferation, considering the basal point for each cell line (NTC clones) as 100%. The plots represent mean ± SEM of assays performed in triplicate. NTC clones were used as reference for statistical calculations. An unpaired t-test was applied for statistical analysis; * *p* < 0.05.

**Figure 5 ijms-23-08463-f005:**
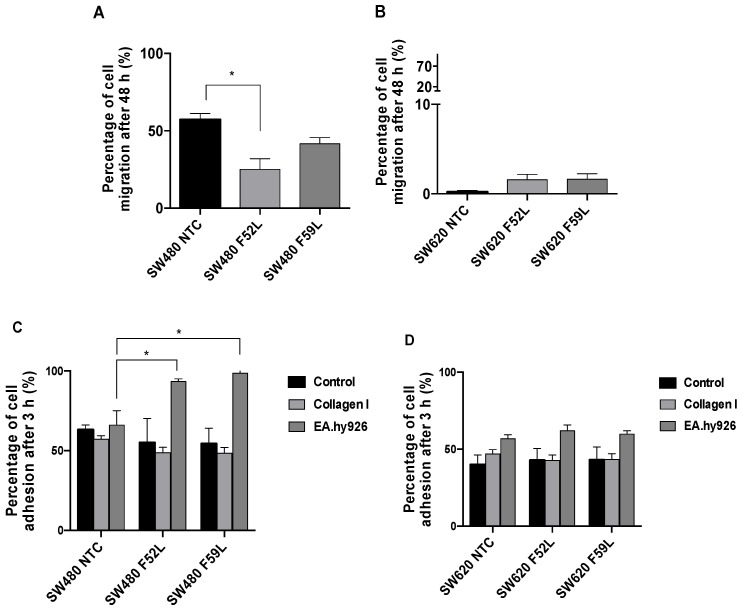
Migration and adhesion capacity of SW480 and SW620 *FUT8* knockdown cells. (**A**,**B**): Transwell Boyden chamber assays interrogated the cellular migration of *FUT8*-silenced F52 and F59 clones compared to respective non-targeted controls. The cells were loaded in the upper side of the Boyden chambers, and incubated for 48 h. Migratory cells passing through the membrane were collected after trypsin detaching, and stained with calcein, and their fluorescence intensity was measured. The experiments were performed three times. The cells were seeded in triplicate, and the results were plotted as percentages of the migrated cells relative to the seeded cells. (**C**,**D**): Adhesion of the SW480 and SW620 NTC cells and the *FUT8*-silenced counterparts was tested using conventional culture plates, collagen type I-coated plates, and plates with a monolayer of EA.hy926 cells mimicking an endothelial barrier. Adhesive capacity was measured 3 h after seeding. The results were plotted as percentages of the adhered cells relative to the initial seeding number. Plots represent mean ± SEM of quadruplicate wells from three experiments. NTC clones were used as reference for statistical calculations. An unpaired *t*-test was applied for statistical analysis; * *p* < 0.05.

## Data Availability

Not applicable.

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
