# Peer review of "Inhibition of α(1,6)fucosyltransferase: Effects on Cell Proliferation, Migration, and Adhesion in an SW480/SW620 Syngeneic Colorectal Cancer Model"

_ijms, 2022, doi:10.3390/ijms23158463_

Round 1

Reviewer 1 Report

Hypothesis and aim of study are clearly presented. The article is well written and data are supported by test anf figures. Conclusions are adherent with the goals.

Author Response

The authors deeply appreciate the positive assessment that this expert makes of our work.

Reviewer 2 Report

thank you for allowing me to review this paper. in the introduction the authors talk about the difficulties of early diagnosis of colorectal cancer and that population-based screening programmes only capture a small percentage of patients. to our knowledge, in all countries where a population-based screening programme has been implemented, the prognosis of colorectal cancer has been significantly improved due to a significant increase in incidence and a significant reduction in mortality. the problem is not the screening programme itself but the low participation which according to the recommendations should be 45%. a clarification seems necessary.

in the introduction the authors report that increased alpha FT activity in the tumour tissue was correlated with the stage of the tumour and lymph node invasion. in the next sentence the authors state that this alpha FT activity was inversely correlated with the degree of infiltration. to our knowledge there is a contradiction here because the greater the degree of tumour infiltration, the greater the risk of lymph node invasion and metastases. can the authors clarify?

We also suggest that the authors revise the plan of the discussion with a first chapter summarising all the results of their study and then discuss the strengths and weaknesses in order to nuance the conclusions and provide perspectives.   indeed, although this is a basic research study, it is difficult to identify the potential clinical impact in colorectal cancer.

Author Response

The authors would like to sincerely thank the Referee 2 for his comments and indications to improve the manuscript. 

Reviewer 3 Report

In the original publication, the authors investigated the effect of alpha-1,6-FT enzyme's complex inhibition on premetastatic and metastatic CRC cell lines. Their results clearly highlight that core fucosylation is a relevant pathogenetic factor in the cellular model of CRC. Inhibition of the enzyme affects both the EMT process and the phenotypic properties of cancer cells, affecting progression.

The study is well designed and well presented.

The results are clear and support the conclusions of the study. 

The used methods are all adequate, their description may help the third part to repeat the experiments. 

The English language needs minor polishing. 

Also in line 173, SW420 must be corrected to SW480. 

I suggest accepting the manuscript for publication after minor revision.

Author Response

We sincerely appreciate the positive feedback our work has received from the Referee 3. The misspelling of SW480 line, which he has kindly pointed out to us, has been corrected in line 187 of the revised version of the manuscript.

Round 2

Reviewer 2 Report

I thank the authors who responded point by point to the comments.